# Hybrid Bi-Level Index for Shortest Paths in Temporal Networks

**Vasilev M.**
Moscow State University
`s02230030@gse.cs.msu.ru`

## Abstract

Temporal graphs provide a natural model for dynamic relational data arising in modern AI systems, including event streams, temporal knowledge graphs, interaction networks, and transaction systems. Efficient reachability querying in such graphs constitutes a fundamental operation underlying temporal reasoning, feature extraction, and dynamic graph learning.

In this paper, we propose a parameterized hybrid indexing framework for temporal reachability queries. Vertices are adaptively partitioned into two classes depending on the size of their reachable sets, enabling a controllable trade-off between memory usage and query time. Assuming a power-law degree distribution, we derive an analytical model for the proportion of promoted (large) vertices as a function of a promotion threshold. Closed-form asymptotic estimates for memory consumption and expected query time are obtained. We further prove the existence of a unique optimal threshold minimizing a combined memory–time cost functional.

Theoretical predictions are validated experimentally, revealing a characteristic U-shaped dependence of query time on the promotion parameter. The results provide a mathematically grounded foundation for adaptive indexing in large-scale temporal graph analytics and AI-driven dynamic data systems.

## 1 Introduction

Temporal graphs model systems in which interactions occur at specific time instances. Such structures naturally arise in communication networks, social interactions, financial transaction streams, logistics systems, and event-based knowledge graphs Wu et al. (2014); Casteigts et al. (2024). In contrast to static graphs, the validity of a path depends not only on connectivity but also on temporal consistency of edge timestamps. This additional constraint fundamentally alters algorithmic complexity and limits the direct applicability of classical graph algorithms Casteigts et al. (2021); Bumpus & Meeks (2022).

Beyond traditional network analysis, temporal graphs have become increasingly important in AI applications. Modern learning systems operate on dynamic relational data, including temporal knowledge graphs, streaming interaction networks, and evolving dependency structures. Operations such as temporal reachability, time-respecting path extraction, and dynamic influence propagation form essential building blocks for representation learning, temporal reasoning, and graph-based inference pipelines. Consequently, scalable temporal reachability indexing plays a critical role in AI infrastructures processing large-scale event data.

A substantial body of work addresses path and reachability problems in temporal graphs. Wu et al. Wu et al. (2014) formalize earliest-arrival and fastest-path problems, demonstrating that temporal constraints significantly complicate classical formulations. Casteigts et al. Casteigts et al. (2024) analyze different semantics of temporal reachability (strict, simple, proper, happy), highlighting the necessity of precise formal models. Further extensions incorporate waiting-time constraints and structural restrictions Casteigts et al. (2021); Bumpus & Meeks (2022). Large-scale and distributed solutions have also been proposed Zhang et al. (2019); Chen et al. (2019); Ding et al. (2021).

For static graphs, reachability indexing via labeling schemes has proven highly effective. Notable examples include 2-hop labeling methods Cohen et al. (2003), hub-based labeling techniques Abraham et al. (2010), and scalable label construction algorithms Ding et al. (2008). However, incorporating temporal constraints into such frameworks is nontrivial due to the combinatorial explosion of time-respecting paths.

An additional challenge arises from the structural properties of real-world networks. Empirical studies show that many large-scale graphs follow power-law degree distributions Barabási & Albert (1999); Clauset et al. (2009). The presence of highly connected hubs leads to strong heterogeneity in reachable-set sizes and temporal influence Kim & Anderson (2011). Ignoring this structural imbalance may result in inefficient memory usage or degraded query performance.

In this work, we introduce a hybrid indexing scheme that explicitly exploits structural heterogeneity. Vertices are partitioned according to a promotion threshold that depends on the size of their reachable sets. Small vertices store precomputed reachability information, while large vertices rely on bounded exploration at query time. The promotion parameter serves as a tunable control governing the trade-off between memory consumption and query latency.

Our contributions are as follows:

- We derive an analytical expression for the fraction of promoted vertices under a power-law degree model.
- We obtain asymptotic estimates for memory usage and expected query time as explicit functions of the promotion threshold.
- We prove the existence of a unique optimal threshold minimizing a combined memory–time cost functional.
- We empirically validate the theoretical predictions and explain the observed U-shaped behavior of query time through architectural effects.

The proposed framework provides a mathematically grounded perspective on adaptive indexing in temporal graphs. It bridges structural graph theory, asymptotic analysis, and scalable algorithm design, and is directly applicable to AI systems operating on dynamic relational data.

## 2 TEMPORAL GRAPH MODEL

We consider a directed *temporal graph*

$$G = (V, E),$$

where each edge is represented by a time-stamped event

$$(u, v, t),$$

indicating that a transition from vertex $u$ to vertex $v$ is possible at time $t$ Wu et al. (2014); Casteigts et al. (2024; 2021). We assume that events arrive in non-decreasing order of time, which reflects streaming dynamic systems in which the chronological order of interactions is essential Bumpus & Meeks (2022).

A temporal (time-respecting) path is a sequence of edges

$$(u_0, u_1, t_1), (u_1, u_2, t_2), \ldots, (u_{k-1}, u_k, t_k)$$

such that

$$t_1 \leq t_2 \leq \cdots \leq t_k.$$

The earliest-arrival distance from $u$ to $v$ is the minimum time $t$ such that $v$ is reachable from $u$ by a temporal path arriving at time $t$ Wu et al. (2014). This metric plays a central role in temporal reachability analysis and dynamic path planning.

Computing such paths efficiently is nontrivial. Wu et al. Wu et al. (2014) study earliest-arrival and fastest paths, demonstrating that straightforward adaptations of static graph traversal are often inefficient in high-volume event streams. Distributed solutions for large-scale temporal graphs have been

proposed in Zhang et al. (2019), while labeling and hub-based indexing techniques have proven effective in static settings Ding et al. (2008); Cohen et al. (2003); Abraham et al. (2010). However, extending these approaches to temporal graphs requires explicit handling of time constraints, significantly increasing computational and storage costs.

Real-world networks frequently exhibit heavy-tailed degree distributions Barabási & Albert (1999); Clauset et al. (2009). Such heterogeneity leads to large variability in reachable-set sizes and temporal influence Kim & Anderson (2011). Therefore, indexing strategies must account for structural imbalance in order to achieve scalable performance.

The adopted model captures the essential aspects of temporal dynamics and serves as a foundation for constructing adaptive hybrid indices for reachability queries in large, structurally heterogeneous networks.

## 3 HYBRID TEMPORAL REACHABILITY INDEX

### 3.1 PROBLEM FORMULATION

Given a stream of temporal edges, we aim to maintain a data structure that supports efficient earliest-arrival distance queries between vertices. A temporal path is valid only if timestamps along the path are non-decreasing Wu et al. (2014); Casteigts et al. (2021).

Classical indexing approaches such as 2-hop labeling Cohen et al. (2003), hub-based labeling Abraham et al. (2010), and distributed reachability frameworks Zhang et al. (2019) achieve strong performance in static graphs. However, their direct application to dynamic temporal graphs is hindered by high storage overhead and the need to encode temporal constraints explicitly.

### 3.2 MAIN IDEA

The key observation is that vertices differ significantly in the size of their reachable predecessor sets Kim & Anderson (2011); Bumpus & Meeks (2022). This structural heterogeneity motivates an adaptive strategy:

- For vertices with small reachable sets, it is efficient to store complete reachability tables.
- For vertices with large reachable sets, full storage becomes memory-intensive and unnecessary.

Accordingly, vertices are partitioned into two classes: **small** and **large**. The boundary between them is determined by a promotion threshold $B$.

### 3.3 STORED DATA STRUCTURES

For each vertex $v$, we maintain:

- direct_edges($v$) — the set of incoming temporal edges;
- dist_all($v$) — a mapping to the vertices with the time of the earliest arrival, representing complete information about reachability Ding et al. (2008); Cohen et al. (2003).

For **small** vertices, both dist_all($v$) and direct_edges($v$) are stored. For **large** vertices, only direct_edges($v$) are retained. Thus, large vertices keep only local structural information, reducing memory consumption.

### 3.4 EVENT PROCESSING

Upon receiving a new event

$$(u \rightarrow v, t),$$

the algorithm performs the following steps:

1. Insert the edge into direct_edges($v$).

2. If $u$ is small, then for each $x \in \text{dist\_all}(u)$, check temporal consistency and update $\text{dist\_all}(v)$ accordingly.

After updating, the promotion condition is evaluated:

$$|\text{dist\_all}(v)| \geq B.$$

If satisfied, vertex $v$ is promoted to the large class, and its reachability table is discarded.

---

**Algorithm 1** Processing a Temporal Edge

---

**Require:** Temporal edge $(u, v, t)$
 1: Insert $(u, t)$ into direct_edges$(v)$
 2: **if** $v$ is small **then**
 3:   Update dist_all$(v)$ with $(u, t)$
 4:   **if** $u$ is small **then**
 5:     **for** each $(x, t_x) \in \text{dist\_all}(u)$ **do**
 6:       **if** $t_x \leq t$ **then**
 7:         update $(x, t)$ in dist_all$(v)$
 8:       **end if**
 9:     **end for**
10:   **end if**
11: **end if**
12: **if** $|\text{dist\_all}(v)| \geq B$ **then**
13:   promote $v$ to large
14: **end if**

---

### 3.5 DISTANCE QUERY

For a query $(s, t)$, two cases arise.

**Case 1: $t$ is small.** The answer is obtained directly:

$$\text{return dist\_all}(t)[s].$$

This requires $O(1)$ time.

**Case 2: $t$ is large.** A backward traversal over incoming edges is performed:

- Only edges respecting temporal ordering are explored Casteigts et al. (2021); Bumpus & Meeks (2022).
- Upon reaching a small vertex, its precomputed reachability table is used.

Thus, large vertices are processed lazily, while small vertices provide constant-time lookups.

### 3.6 HYBRID TRADE-OFF

The method combines two operational regimes:

| Vertex type | Memory usage | Query time |
|:---:|:---:|:---:|
| Small | High | $O(1)$ |
| Large | Low | Graph traversal |

The threshold $B$ controls the trade-off:

$$\text{memory} \longleftrightarrow \text{query latency}.$$

### 3.7 Structural Interpretation

The algorithm may be viewed as an adaptive reachability index:

- Locally sparse regions are collapsed into reachability tables.
- Globally dense or hub-like vertices remain explicit.

This adaptive separation aligns naturally with power-law degree distributions Barabási & Albert (1999); Clauset et al. (2009). In practice, large vertices tend to coincide with structural hubs, providing automatic detection of high-influence nodes without explicit preprocessing.

The approach can be interpreted as a hybridization of hub-based labeling Abraham et al. (2010), 2-hop labeling Cohen et al. (2003), and lazy exploration strategies Bumpus & Meeks (2022), while remaining fully adaptive to the input temporal graph.

## 4 Degree Distribution

We assume that the in-degree distribution of vertices follows a *power-law* (heavy-tailed) distribution Barabási & Albert (1999); Clauset et al. (2009):

$$\Pr[K \geq k] = \left(\frac{k}{k_{\min}}\right)^{1-\gamma}, \qquad \gamma > 1.$$

Such distributions are characteristic of many real-world networks, including social, communication, and transportation systems, where a small number of hub vertices possess very high degree, while the majority of vertices have low degree Kim & Anderson (2011).

In the context of temporal graphs, this heterogeneity explains the observed variability in reachable-set sizes:

- **Small vertices** with low degree typically have limited reachable sets and can efficiently store complete reachability tables;
- **Large vertices** with high degree act as structural hubs, strongly influencing reachability flow, and are more efficiently processed lazily to reduce memory usage Bumpus & Meeks (2022).

Therefore, the power-law degree distribution plays a central role in the construction of the hybrid index, as it enables analytical estimation of the fraction of small and large vertices and prediction of memory and query-time behavior.

## 5 Fraction of Large Vertices

Let $K$ denote the in-degree of a vertex. Assume that the degree distribution has a power-law tail:

$$P(K = k) = Ck^{-\gamma}, \qquad k \geq k_{\min}, \quad \gamma > 1.$$

### 5.1 Normalization

The normalization constant $C$ is determined by

$$\sum_{k=k_{\min}}^{\infty} Ck^{-\gamma} = 1.$$

Using a continuous approximation, we replace the sum with an integral:

$$\int_{k_{\min}}^{\infty} Ck^{-\gamma} \, dk = 1.$$

Evaluating the integral,

$$C \int_{k_{\min}}^{\infty} k^{-\gamma}\, dk = C \left[ \frac{k^{1-\gamma}}{1-\gamma} \right]_{k_{\min}}^{\infty}.$$

Since $\gamma > 1$, the upper limit vanishes, yielding

$$C \frac{k_{\min}^{1-\gamma}}{\gamma - 1} = 1.$$

Hence,

$$\boxed{C = (\gamma - 1)k_{\min}^{\gamma-1}.}$$

## 5.2 Probability of Exceeding a Degree Threshold

The fraction of large vertices is defined as the probability that the degree exceeds a promotion threshold $k_p(B)$:

$$p_L(B) = \Pr[K \geq k_p(B)].$$

Using the continuous approximation,

$$p_L(B) = \int_{k_p(B)}^{\infty} Ck^{-\gamma}\, dk.$$

Substituting the normalization constant,

$$p_L(B) = (\gamma - 1)k_{\min}^{\gamma-1} \int_{k_p(B)}^{\infty} k^{-\gamma}\, dk.$$

Evaluating the integral,

$$\int_{k_p(B)}^{\infty} k^{-\gamma}\, dk = \frac{k_p(B)^{1-\gamma}}{\gamma - 1}.$$

After cancellation, we obtain

$$\boxed{p_L(B) = \left( \frac{k_p(B)}{k_{\min}} \right)^{1-\gamma}.}$$

## 5.3 Relation Between Degree Threshold and Promotion Parameter

Assume that the expected size of the reachable set of a vertex with degree $k$ scales as a power function:

$$\mathbb{E}[R_{\text{in}}(k)] = C_R k^{\delta}.$$

The boundary between small and large vertices is defined by

$$C_R k_p^{\delta} = B.$$

Solving for $k_p(B)$ yields

$$k_p(B) = \left( \frac{B}{C_R} \right)^{1/\delta}.$$

## 5.4 Final Expression

Substituting $k_p(B)$ into the expression for $p_L(B)$, we obtain

$$p_L(B) = \left( \frac{(B/C_R)^{1/\delta}}{k_{\min}} \right)^{1-\gamma}.$$

Rearranging terms,

$$p_L(B) = k_{\min}^{\gamma-1} C_R^{\frac{\gamma-1}{\delta}} B^{\frac{1-\gamma}{\delta}}.$$

Therefore,

$$\boxed{p_L(B) = A\, B^{\frac{1-\gamma}{\delta}}}$$

where

$$\boxed{A = k_{\min}^{\gamma-1} C_R^{\frac{\gamma-1}{\delta}}.}$$

## 5.5 LOGARITHMIC FORM

Taking logarithms,

$$\ln p_L = \frac{1-\gamma}{\delta}\ln B + \ln A.$$

Thus, the relationship is linear in log–log coordinates, with slope

$$\boxed{\alpha = \frac{1-\gamma}{\delta}.}$$

This power-law dependence enables empirical validation through log–log regression and provides a predictive model for how the fraction of large vertices decreases as the promotion threshold increases. (see Fig. 1).

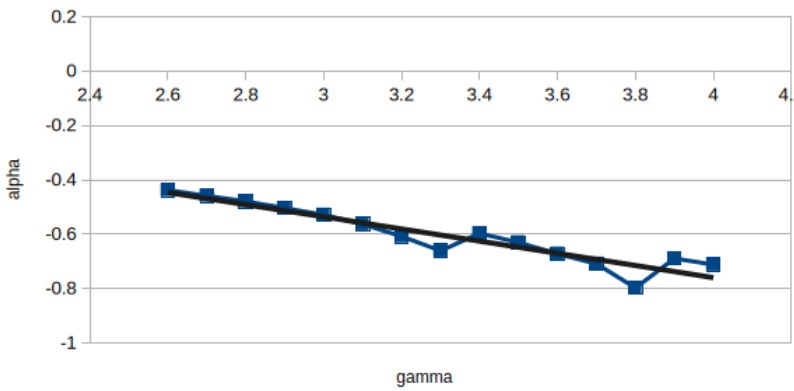

Figure 1: Dependence of the slope of a logarithmic graph on gamma

## 6 MEMORY MODEL

Let $n = |V|$ denote the number of vertices and $m = |E|$ the number of temporal edges (events).

Storing the underlying temporal graph requires

$$M_{\text{edges}} = O(m).$$

Additional memory is required to store reachability structures for small vertices.

Let $p_L(B)$ denote the fraction of large vertices under promotion threshold $B$. Then the fraction of small vertices equals $1 - p_L(B)$.

Each small vertex stores at most $B$ reachable sources. Therefore, the total memory required for reachability structures is

$$M_{\text{reach}} = O\big((1 - p_L(B))\, n\, B\big).$$

Hence, the overall memory consumption is

$$\boxed{M(B) = O\big(m + (1 - p_L(B))\, n\, B\big).}$$

## 6.1 Asymptotic Form

Substituting the model for the fraction of large vertices

$$p_L(B) = A\, B^{\frac{1-\gamma}{\delta}},$$

we obtain

$$M(B) = O\left(m + nB - nAB^{1+\frac{1-\gamma}{\delta}}\right).$$

Let

$$\eta = 1 + \frac{1-\gamma}{\delta}.$$

Then

$$M(B) = O\big(m + nB - nAB^{\eta}\big).$$

## 6.2 Shape of the Memory Curve

The function $M(B)$ is the difference between two increasing terms:

- a linear growth term $nB$,
- a sublinear power term $B^{\eta}$.

Under typical heavy-tailed degree parameters,

$$0 < \eta < 1,$$

so for sufficiently large $B$, the linear term dominates:

$$M(B) \sim nB.$$

Thus, memory grows approximately linearly in the high-$B$ regime.

Empirically, the observed curve exhibits a convex, "right-branch-of-a-parabola" shape: initially almost flat for small $B$, followed by nearly linear growth for larger thresholds (see Fig. 2).

## 6.3 Extreme Regimes

**Small $B$.** When the threshold is small, almost all vertices become large:

$$p_L(B) \approx 1.$$

Dependence of memory on promotion thresholds B

Figure 2: Dependence of memory on promotion thresholds B with $\gamma = 3.5$

Thus,

$$M(B) \approx O(m),$$

and memory becomes nearly independent of $B$.

**Large $B$.** When the threshold is large, most vertices are small:

$$p_L(B) \approx 0.$$

Then

$$M(B) \approx O(m + nB),$$

leading to approximately linear growth.

## 6.4  PRACTICAL IMPLICATION

Memory is monotonically increasing in $B$. Therefore, the optimal threshold must arise from a trade-off between memory growth and query-time behavior.

## 7  QUERY TIME MODEL

We analyze the expected query response time as a function of the promotion threshold $B$.

Queries are processed differently depending on the vertex type:

- For **small** vertices, the answer is retrieved from the precomputed reachability table;
- For **large** vertices, a partial graph traversal is performed.

## 7.1  BASELINE MODEL

Let

$$T_s = O(1), \qquad T_\ell = O(\tau),$$

where $T_s$ denotes the average cost of a small-vertex query, and $T_\ell$ the average traversal cost for large vertices.

If $p_L(B)$ denotes the fraction of large vertices, then the expected query time equals

$$T(B) = (1 - p_L(B))T_s + p_L(B)T_\ell.$$

Substituting the power-law model

$$p_L(B) = A\,B^{\frac{1-\gamma}{\delta}},$$

we obtain

$$T(B) = T_s + (T_\ell - T_s)\,A\,B^{\frac{1-\gamma}{\delta}}.$$

Since $\gamma > 1$, the exponent is negative, and the baseline model predicts monotonic decay of query time as $B$ increases.

## 7.2 REFINED MODEL

Empirical results demonstrate instead a U-shaped dependence.(see Fig. 3).

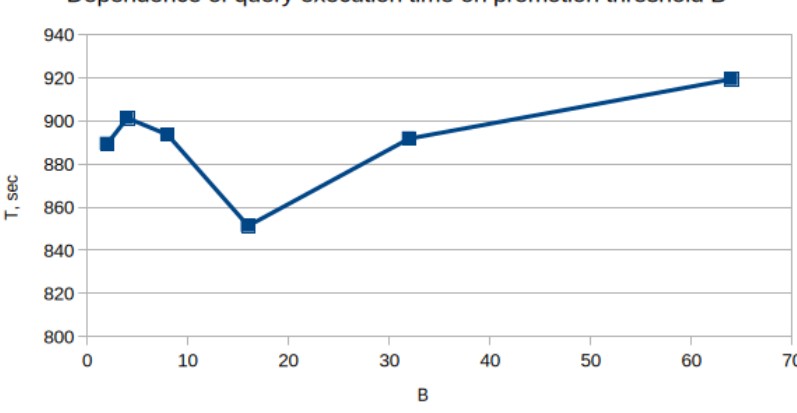

Figure 3: Dependence of query execution time on promotion threshold B

The discrepancy arises because $T_s$ is not constant in practice. Reachability tables are implemented using hash tables (`unordered_map` and Robin Hood hashing), which provide $O(1)$ average complexity, but actual performance depends on:

- load factor,
- probe length,
- cache locality,
- memory hierarchy effects.

As $B$ increases, table sizes scale approximately as $nB$, which degrades cache locality and increases average lookup latency.

Thus,

$$T_s = T_s(B), \qquad \frac{dT_s}{dB} > 0.$$

The refined model becomes

$$T(B) = (1 - p_L(B)) \, T_s(B) + p_L(B) \, T_\ell(B).$$

## 8    OPTIMAL THRESHOLD

Empirical observations indicate that $T(B)$ exhibits a U-shaped dependence on $B$.

We model query time as

$$T(B) = aB^{-\beta} + cB^\theta, \qquad \beta > 0, \ \theta > 0,$$

where the first term captures reduction in large-vertex traversals, and the second term models increasing hash-table lookup cost.

Memory scales approximately as

$$M(B) \sim dB, \qquad d > 0.$$

### 8.1    COMBINED COST FUNCTION

Define

$$C(B) = w_t T(B) + w_m M(B), \qquad w_t, w_m > 0.$$

Substituting,

$$C(B) = aB^{-\beta} + cB^\theta + dB.$$

### 8.2    EXISTENCE OF A MINIMUM

We have

$$\lim_{B \to 0} C(B) = \infty, \qquad \lim_{B \to \infty} C(B) = \infty.$$

Thus, by continuity, a global minimum exists.

Under typical parameter values, the derivative

$$C'(B) = -a\beta B^{-\beta-1} + c\theta B^{\theta-1} + d$$

is strictly increasing, implying uniqueness of the minimizer.

## 9    CONCLUSION

We investigated a parameterized hybrid indexing algorithm for temporal reachability queries, controlled by a promotion threshold $B$.

Theoretical analysis showed that the fraction of large vertices decreases as a power law in $B$, while memory grows approximately linearly. A simplified model predicted monotonic improvement in query time.

However, empirical evaluation revealed a U-shaped behavior. We demonstrated that this phenomenon is explained by non-asymptotic effects of hash-table implementations, including cache locality degradation and memory hierarchy costs.

As a result, an internal optimal threshold $B^*$ exists, minimizing average query time.

The threshold parameter therefore acts as a tunable mechanism for balancing memory consumption and runtime performance in large-scale temporal graphs.

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
