# OpenReview forum: "Hybrid Bi-Level Index for Shortest Paths in Temporal Networks"
_mathai.club/MathAI/2026/Conference — 2026 Oral_

### Official Review · Reviewer_Xjgv · 2026-03-11
**Interesting hybrid index for temporal reachability with clean theory, but relies on strong assumptions and has limited empirical validation.**

**Rating:** 7
**Confidence:** 3

**Review:**

# Quality
The paper presents a conceptually simple hybrid index that partitions vertices into “small” and “large” based on a promotion threshold, and analyzes how this choice affects memory consumption and query time in temporal graphs. The theoretical modeling under power-law assumptions is technically sound and internally consistent, and the high-level algorithmic idea is reasonable, but important implementation details and complexity guarantees remain at a fairly abstract level, which makes it harder to fully assess robustness and worst-case behavior.

# Clarity
The exposition is generally clear and logically structured: the temporal-graph model, the hybrid indexing scheme, and the analytic derivations are explained in an accessible way. However, some references to empirical results (for example, a figure that is supposed to illustrate the U-shaped dependence of query time on the promotion parameter) are present, but the actual figure seems to be missing, and the description of experimental settings and baselines is not sufficiently detailed to fully understand the practical impact.

# Originality
The work combines existing ideas from reachability indexing, hub-based and 2-hop labeling, and temporal-graph analysis into a bi-level scheme tailored to temporal reachability. The explicit analytical link between the promotion threshold and the fraction of “large” vertices under power-law assumptions is a nice conceptual contribution, even if the overall approach feels more incremental than radically new.

# Significance
Temporal reachability is an important primitive for large-scale dynamic data and AI systems, and having a tunable index with a mathematically interpretable parameter is practically useful. At the same time, the current empirical evidence is somewhat limited in scope, and the absence of robustness or worst-case analysis makes it harder to judge the method’s reliability in more challenging or adversarial scenarios, which reduces the overall impact.

# Pros
- Clear and intuitive hybrid indexing idea with a single promotion threshold controlling the memory–latency trade-off.
- Clean and coherent analytical treatment under power-law assumptions, leading to interpretable scaling relations.
- Honest attempt to account for systems-level effects (hash tables, cache locality) rather than relying only on idealized models.
- Good contextualization within temporal-graph and reachability-indexing literature.

# Cons
- Strong reliance on specific structural assumptions (power-law degrees, simple scaling of reachable sets) with little analysis of sensitivity to their violation.
- Lack of worst-case or robustness discussion; behavior under adversarial or pathological temporal dynamics is not addressed.
- Algorithmic description and complexity guarantees are high-level, which hurts reproducibility and rigorous performance assessment.
- The text explicitly refers to a figure illustrating the U-shaped query-time behavior, but this figure is not actually present in the manuscript, which weakens the empirical argument and makes that part of the explanation hard to verify.

---

### Official Review · Reviewer_ubDH · 2026-03-13
**The idea of ​​the study is interesting, but the text of the paper is not finished.**

**Rating:** 4
**Confidence:** 4

**Review:**

This paper addresses problem of efficient reachability querying in temporal graphs. The authors propose a parameterized, adaptive indexing framework that partitions vertices into "small" and "large" classes based on the size of their reachable sets, allowing for a tunable trade-off between memory consumption and query time. The work is grounded in an analytical model that assumes a power-law degree distribution, leading to closed-form estimates for memory usage and query time, and a proof for the existence of an optimal threshold. While the core idea is clear and promising, the paper in its current form is significantly underdeveloped and reads more like an extended abstract than a complete conference submission.
In particular, Section 3 lacks the main algorithm. Beginning with Section 5, the paper is a draft with a number of formulas requiring explanatory text. Figures are missing, although references to them are found in the text (e.g. line 381). Section 8.3 consists of 9 words.

---

### Official Review · Reviewer_pMyu · 2026-03-13
**Promising Hybrid Indexing Approach with Some Presentation Gaps**

**Rating:** 6
**Confidence:** 4

**Review:**

This paper studies efficient querying in temporal graphs through a hybrid bi-level indexing strategy that partitions vertices into “small” and “large” classes according to a promotion threshold.
The main idea is intuitive: small vertices store full reachability information, while large vertices are handled more lazily at query time, yielding a tunable trade-off between memory usage and latency. The paper also develops an analytical model under power-law assumptions and argues for the existence of an optimal threshold balancing query time and memory.
Overall, I found the paper interesting and reasonably well motivated, and I believe it has enough merit to be considered slightly above the acceptance threshold, although the current version still has several presentation and validation issues.

A key strength of the paper is that the proposed indexing idea is simple, interpretable, and practically meaningful. The threshold-based partitioning gives a clean way to control the memory–time trade-off, and the structural intuition behind separating low-reachability and hub-like vertices is convincing. I also appreciated the effort to connect the algorithmic design with a power-law model of degree heterogeneity and to derive explicit asymptotic expressions for the fraction of promoted vertices, memory usage, and expected query time. This gives the method a clearer analytical foundation than many purely heuristic indexing schemes.

However, the paper also has several weaknesses that prevent it from being a clearly strong acceptance.
First, the manuscript is more convincing at the conceptual and analytical level than at the algorithmic and experimental level. Section 3 gives the main idea and the data structures, but the full algorithmic specification is still fairly high-level, making it difficult to assess reproducibility, update complexity, and worst-case behavior in detail.
Second, the empirical validation is not yet presented in a sufficiently complete way: the paper refers to experimental evidence and to a figure illustrating the claimed behavior, but the manuscript still contains “Fig. ??” (Section 6.2) rather than an actual figure. This weakens the practical case considerably.

Overall, my view is positive but cautious. The paper contains a solid and interesting idea, and the analytical treatment is coherent enough that I can see value in accepting it. At the same time, I would strongly encourage the authors to improve the completeness of the algorithm description, clarify the exact problem formulation, and substantially strengthen the empirical section in the final version.

---

### Decision · Program_Chairs · 2026-03-14

**Decision:**

Accept (Oral)

**Comment:**

Dear Author(s),

On behalf of the Program Committee of the International Conference on Mathematics of Artificial Intelligence (MathAI 2026), we are pleased to inform you that your paper has been accepted for an oral presentation at MathAI 2026.

Your paper was evaluated through a rigorous two-stage review process involving both automated screening and expert review by members of the Program Committee. The reviewers recognized the quality and contribution of your work.

Presentation details:

- Format: Oral presentation (15–20 minutes + 5 minutes Q&A)
- Mode: You may present either in person (offline) at the conference venue in Sirius, Russia, or remotely via Zoom. Please indicate your preferred mode when confirming your participation.
- Conference dates: Marh 30 - April 3, 2026
- Website: https://mathai.club

Next steps:

1. Please confirm your participation and presentation mode by replying to this email mathai.club@yandex.ru no later than March 15, 2026 18:00 Moscow time.
2. If you plan to attend in person, the organizing committee will provide accommodation details separately.
3. Please prepare your final camera-ready manuscript according to the formatting guidelines available at https://mathai.club and upload it to OpenReview by March 15, 2026 18:00 Moscow time.

Should you have any questions regarding the program, logistics, or your presentation slot, please do not hesitate to contact us.

We look forward to your contribution to MathAI 2026.

With kind regards,

MathAI 2026 Program Committee
International Conference on Mathematics of Artificial Intelligence
https://mathai.club
OpenReview: https://openreview.net/group?id=mathai.club/MathAI/2026/Conference
Telegram: https://t.me/MathAI_club
Email: mathai.club@yandex.ru